# Genetic Deletion of *Mmp9* Does Not Reduce Airway Inflammation and Structural Lung Damage in Mice with Cystic Fibrosis-like Lung Disease

**DOI:** 10.3390/ijms232113405

**Published:** 2022-11-02

**Authors:** Claudius Wagner, Anita Balázs, Jolanthe Schatterny, Zhe Zhou-Suckow, Julia Duerr, Carsten Schultz, Marcus A. Mall

**Affiliations:** 1Department of Translational Pulmonology, University of Heidelberg, 69117 Heidelberg, Germany; 2Translational Lung Research Center (TLRC), Member of the German Center for Lung Research (DZL), Im Neuenheimer Feld 156, 69120 Heidelberg, Germany; 3Department of Pediatric Respiratory Medicine, Immunology and Critical Care Medicine, Charité—Universitätsmedizin Berlin, Augustenburger Platz 1, 13353 Berlin, Germany; 4Berlin Institute of Health, Charité—Universitätsmedizin Berlin, Charitéplatz 1, 10117 Berlin, Germany; 5Department of Chemical Physiology and Biochemistry, Oregon Health & Science University, Portland, OR 97239, USA; 6German Center for Lung Research (DZL), Associated Partner Site, Augustenburger Platz 1, 13353 Berlin, Germany

**Keywords:** matrix metalloproteinase 9, neutrophil elastase, airway inflammation, lung damage

## Abstract

Elevated levels of matrix metalloprotease 9 (MMP-9) and neutrophil elastase (NE) are associated with bronchiectasis and lung function decline in patients with cystic fibrosis (CF). MMP-9 is a potent extracellular matrix-degrading enzyme which is activated by NE and has been implicated in structural lung damage in CF. However, the role of MMP-9 in the in vivo pathogenesis of CF lung disease is not well understood. Therefore, we used β-epithelial Na^+^ channel-overexpressing transgenic (βENaC-Tg) mice as a model of CF-like lung disease and determined the effect of genetic deletion of *Mmp9* (*Mmp9*^-/-^) on key aspects of the pulmonary phenotype. We found that MMP-9 levels were elevated in the lungs of βENaC-Tg mice compared with wild-type littermates. Deletion of *Mmp9* had no effect on spontaneous mortality, inflammatory markers in bronchoalveolar lavage, goblet cell metaplasia, mucus hypersecretion and emphysema-like structural lung damage, while it partially reduced mucus obstruction in βENaC-Tg mice. Further, lack of *Mmp9* had no effect on increased inspiratory capacity and increased lung compliance in βENaC-Tg mice, whereas both lung function parameters were improved with genetic deletion of *NE*. We conclude that MMP-9 does not play a major role in the in vivo pathogenesis of CF-like lung disease in mice.

## 1. Introduction

Chronic neutrophilic inflammation resulting in increased protease activity plays a key role in the progression of chronic lung disease in patients with cystic fibrosis (CF) [1,2]. Upon recruitment to the CF airways, activated neutrophils release their granules containing proteolytic enzymes, such as neutrophil elastase (NE) and matrix-metalloproteinase 9 (MMP-9), resulting in a high protease burden that overwhelms the anti-protease shield and leads to progressive structural damage and lung function decline [3,4,5]. Free NE activity in the bronchoalveolar lavage (BAL) fluid of CF patients is an established risk factor for bronchiectasis and a series of previous studies demonstrated that ”free” NE activity is a key driver of sustained inflammation, mucus hypersecretion and airway remodeling [6,7,8,9,10,11,12]. 

In addition to NE, emerging evidence suggests that MMP-9, a protease with redundant functions to NE, may also play an important role in the pathogenesis of CF lung disease [13]. MMP-9, also known as gelatinase B, is a potent extracellular matrix (ECM) degrading enzyme, considered as one of the main effector proteases of tissue remodeling in other chronic inflammatory lung diseases such as COPD and asthma [14,15]. Furthermore, ECM cleavage products and cytokines processed by MMP-9 have been implicated in leukocyte infiltration [16]. Although MMP-9 is ubiquitously expressed by a variety of cell types, neutrophils contain substantial amounts of MMP-9 and constitute the primary source in neutrophilic airway inflammation [17]. MMP-9 is secreted as a pro-enzyme, which is activated by NE or potentially other proteases, and MMP-9 activity is neutralized by binding to its endogenous inhibitor, the tissue inhibitor of matrix metalloproteinases 1 (TIMP1) [18]. Biochemical studies showed reciprocal processing of endogenous inhibitors of MMP-9 and NE, where MMP-9 was able to inactivate the NE inhibitor α1-antitrypsin and NE, in turn, inactivated TIMP1 [19,20], suggesting that MMP-9 and NE may reciprocally increase their activities in CF airways. A series of observational studies demonstrated an association between MMP-9 activity in BAL fluid and lung disease severity in patients with CF [21,22,23,24]. In addition, MMP-9 levels in BAL were found to correlate with the development of bronchiectasis in preschool children with CF supporting its role in early disease progression [25]. In that study, MMP-9 activity directly correlated with free NE activity, suggesting a causal link between the two proteases in CF lung disease [25]. 

When viewed in combination, the results of these association studies in CF patients suggest that MMP-9 may be implicated in the pathogenesis of CF lung disease. However, a causal role of MMP-9 and the relative roles of MMP-9 vs. NE in the in vivo pathogenesis of CF lung disease have not been established. The aim of this study was therefore to investigate the in vivo role of MMP-9 in mice with airway-specific overexpression of the β-subunit of the epithelial Na^+^ channel (βENaC-Tg) that exhibit CF-like airway surface dehydration and impaired mucociliary clearance, and develop spontaneous lung disease that shares key features of patients with CF including early onset airway mucus plugging, chronic neutrophilic inflammation and structural lung damage [26,27,28,29]. To achieve this goal, we crossed βENaC-Tg mice with *Mmp9*-deficient (*Mmp9*^-/-^) mice and determined the effect of genetic deletion of *Mmp9* on the pulmonary phenotype of the progeny. Specifically, we monitored survival, measured MMP-9 and other inflammatory markers in BAL fluid, and assessed airway mucus content and structural lung disease in lung sections of wild-type (WT), *Mmp9*^-/-^, βENaC-Tg and βENaC-Tg/*Mmp9*^-/-^ littermates. Further, we compared the effect of *Mmp9* vs. *NE* deficiency on lung function to assess their relative contributions to CF-like lung disease in βENaC-Tg mice.

## 2. Results

### 2.1. MMP-9 Protein Levels ArFe Elevated in BAL Fluid of βENaC-Tg Mice

To study the potential contribution of MMP-9 in the pathogenesis of CF-like lung disease, we assessed levels of secreted MMP-9 protein in BAL supernatant of adult WT, *Mmp9*^-/-^, βENaC-Tg and βENaC-Tg/*Mmp9*^-/-^ mice by gelatin zymography. A predominant band of pro-MMP-9 (~92 kDa) and two different bands of active MMP-9 (~82 kDa and ~75 kDa) were visible in βENaC-Tg mice, but not in WT mice (Figure 1). BAL supernatant of *Mmp9*^-/-^ or βENaC-Tg/*Mmp9*^-/-^ showed no bands of either pro- or active MMP-9. In addition, constitutive expression of pro-MMP-2 (~72 kDa) and MMP-2 (~65 kDa) were observed across all genotypes. Quantitative analysis of total MMP-9 band intensities indicated a strong upregulation of secreted MMP-9 protein levels in βENaC-Tg mice.

### 2.2. Lack of MMP-9 Does Not Reduce Mortality in βENaC-Tg Mice

CF-like lung disease in βENaC-Tg mice is associated with spontaneous mortality in the first weeks of life [30,31,32]. To investigate the effect of *Mmp9* deletion on mortality, we evaluated the genotype distribution of the surviving progeny of the cross of *Mmp9*^+*/-*^ with βENaC-Tg/*Mmp9*^+*/-*^ mice and compared the observed percentage of mice in each of the four possible genotype groups with the percentage of mice expected from the Mendelian ratio (Figure 2). WT and *Mmp9*^-/-^ mice showed normal survival. In line with previous studies, the percentage of surviving βENaC-Tg mice was significantly reduced [29,30,31,32]. A similar mortality was observed in βENaC-Tg/*Mmp9*^-/-^ mice compared to βENaC-Tg mice. 

### 2.3. Genetic Deletion of Mmp9 Does Not Reduce Airway Inflammation in βENaC-Tg Mice 

To evaluate the effect of *Mmp9* deletion on chronic airway inflammation, we determined inflammatory cell counts and cytokine profiles in the BAL fluid of βENaC-Tg and βENaC-Tg/*Mmp9*^-/-^ mice, as well as WT and *Mmp9*^-/-^ littermates (Figure 3). *Mmp9*^-/-^ mice showed similar inflammatory cell counts to WT mice. As expected from previous studies, total cell counts were markedly increased in βENaC-Tg mice compared to WT, which mainly consisted of neutrophils and eosinophils (Figure 3a) [29,30,33]. βENaC-Tg/*Mmp9*^-/-^ mice displayed comparable total and differential cell counts to βENaC-Tg mice. Consistent with previous studies, airway inflammation in the βENaC-Tg mice was associated with significantly increased levels of pro-inflammatory cytokines, including keratinocyte chemoattractant (KC), macrophage inflammatory protein (MIP)-2, Interleukin (IL)-13, IL-1α and IL-1β when compared to WT littermate controls (Figure 3b–f). In βENaC-Tg/*Mmp9*^-/-^ mice, levels of pro-inflammatory cytokines did not differ from βENaC-Tg mice. 

### 2.4. Genetic Deletion of Mmp9 Has No Effect on Goblet Cell Metaplasia and Increased Mucin Expression, but Partially Reduces Mucus Obstruction in βENaC-Tg Mice

Chronic airway inflammation in βENaC-Tg mice is associated with mucus hypersecretion, as evidenced by goblet cell metaplasia and elevated expression of secreted mucins (MUC5AC and MUC5B) that were found to be reduced in βENaC-Tg mice that lack *NE* (βENaC-Tg/*NE*^-/-^) [9]. To determine the effects of genetic deletion of *Mmp9* on mucus hypersecretion, we compared goblet cell counts, expression of goblet cell marker *Gob5*, as well as secreted mucins *Muc5ac* and *Muc5b* in the lungs of βENaC-Tg and βENaC-Tg/*Mmp9*^-/-^ mice and WT and *Mmp9*^-/-^ littermates (Figure 4). These studies showed that goblet cell numbers and mRNA expression of *Gob5*, *Muc5ac* and *Muc5b* were significantly elevated in βENaC-Tg mice compared to WT and *Mmp9*^-/-^ mice. However, there was no difference in goblet cell counts and expression levels of *Gob5*, *Muc5ac*, and *Muc5b* in βENaC-Tg/*Mmp9*^-/-^ mice compared to βENaC-Tg mice. As expected from previous studies [10,27,30], airway sections of βENaC-Tg mice showed accumulation of AB-PAS positive material in the airway lumen, which was absent in WT or *Mmp9*^-/-^ animals (Figure 4f). When compared with βENaC-Tg mice, intraluminal mucus content was partially reduced in βENaC-Tg/*Mmp9*^-/-^ mice.

### 2.5. Genetic Deletion of Mmp9 Does Not Reduce Emphysema-like Structural Lung Damage in βENaC-Tg Mice

Our previous studies showed that chronic airway inflammation in βENaC-Tg mice is associated with emphysema-like structural damage of distal airspaces and that NE and MMP-12 are implicated in this process [9,28,34,35,36]. To determine if MMP-9 contributes to emphysema formation, we performed measurements of lung volume, mean linear intercepts (MLI) and destructive index (DI) in lung tissue sections of βENaC-Tg, βENaC-Tg/*Mmp9*^-/-^ and littermate control mice (Figure 5). Consistent with previous studies [30], alveolar architecture in βENaC-Tg mice appeared abnormal, whereas lung volumes, MLI and DI were significantly elevated compared to those of WT animals. *Mmp9*^-/-^ mice showed comparable alveolar structure, lung volumes and distal airspace morphometry to WT mice. βENaC-Tg/*Mmp9*^-/-^ mice displayed similar emphysematous morphology and no difference in quantitative emphysema parameters compared to βENaC-Tg mice.

### 2.6. Genetic Deletion of NE, but Not Mmp9 Improves Lung Function Impairment in βENaC-Tg Mice

Our previous studies showed that pulmonary emphysema in βENaC-Tg mice is associated with abnormalities in lung function, including a characteristic shift in pressure–volume curves reflecting increased static compliance compared with WT littermates [30,37,38], and that the emphysema-like lung morphology is reduced by genetic deletion of *NE* [9,39]. To test the relative contribution of MMP-9 vs. NE in emphysema formation, we complemented our studies of distal airspace morphometry with invasive pulmonary function testing in βENaC-Tg mice lacking either MMP-9 (βENaC-Tg/*Mmp9*^-/-^) or NE (βENaC-Tg/*NE*^-/-^) (Figure 6). As expected from previous studies, pressure–volume curves were shifted and inspiratory volume and static compliance were significantly increased in the lungs of βENaC-Tg vs. WT mice. Deletion of *Mmp9* had no effect on these lung function parameters in βENaC-Tg mice. In contrast, deletion of *NE* in βENaC-Tg mice resulted in partially restored inspiratory capacity and static compliance toward WT levels.

## 3. Discussion

A series of observational studies found elevated MMP-9 activity in the lungs of patients with CF from early age onward, suggesting MMP-9 as a potential candidate target for anti-protease therapeutic approaches [21,22,23,25]. While previous in vitro studies suggested that NE and MMP-9 may act synergistically in driving progression of structural lung damage in CF [19,25], the in vivo role of MMP-9 in this process is not well understood. We previously found that genetic deletion of *NE* reduces chronic airway inflammation, mucus hypersecretion and structural lung damage in βENaC-Tg mice [9,39]. In contrast, our present study demonstrates that genetic deletion of *Mmp9* has no effect on the development of these key features of CF-like lung disease in βENaC-Tg mice. Further, we found that genetic deletion of *NE*, but not *Mmp9* improves lung function impairment associated with emphysema-like structural lung damage in βENaC-Tg mice. Collectively, our data do not support a major role of MMP-9 in the in vivo pathogenesis of CF-like lung disease in mice. 

First, similar to patients with CF [21,22,23,25], our BAL studies demonstrate that both pro-MMP-9 and active MMP-9 were increased in βENaC-Tg mice, compared to WT mice, confirming the notion that chronic neutrophilic inflammation is associated with elevated MMP-9 in the lung. Despite this upregulation of MMP-9 in βENaC-Tg mice, lack of MMP-9 did not reduce levels of pro-inflammatory cytokines or inflammatory cell counts, indicating that MMP-9 does not contribute to neutrophil recruitment and chronic airway neutrophilia (Figure 3). This is in contrast to a previous study that investigated the role of NE and showed that genetic deletion of *NE* results in a significant reduction in neutrophilic airway inflammation in βENaC-Tg mice [9]. Taken together, these results argue against an important role of MMP-9 activity, as well as synergistic effects of MMP-9 and NE, in the in vivo development and perpetuation of neutrophilic airway inflammation in this model of CF lung disease.

Second, we show that MMP-9 deficiency does not affect goblet cell metaplasia and increased expression of the secreted mucins *Muc5ac* and *Muc5b*, i.e., markers of mucus hypersecretion, in βENaC-Tg mice (Figure 4). Of note, previous studies demonstrated that other proteases released in CF lung disease such as NE and cathepsin S are potent inducers of goblet cell metaplasia and mucin expression, and that genetic ablation of those proteases decreased mucus hypersecretion in the lungs of βENaC-Tg mice [9,40,41]. Although one study implicated MMP-9 in goblet cell hyperplasia in LPS-induced acute lung injury, our findings suggest that MMP-9 does not contribute to mucus hypersecretion associated with chronic inflammation in CF lung disease [42]. Interestingly, we observed a partial reduction of airway mucus obstruction in βENaC-Tg/*Mmp9*^-/-^ mice compared to βENaC-Tg/*Mmp9^+/+^* littermates, suggesting that lack of MMP-9 may improve mucus clearance. However, this speculation including potential mechanisms of the way that lack of MMP-9 may facilitate mucus clearance from the lungs needs to be addressed in future studies. Regardless, lack of MMP-9 had no effect on the spontaneous mortality observed in a subset of βENaC-Tg mice suggesting that the observed reduction in airway mucus content was not sufficient to prevent death due severe mucus plugging (Figure 2). 

Third, we demonstrate by complementary analyses including morphometry of lung tissue and pulmonary function testing that MMP-9 is not implicated in the development of emphysema-like structural lung damage in βENaC-Tg mice (Figure 5 and Figure 6). While MMP-9 is generally believed to be a key player of ECM remodeling, reports are conflicting regarding its role in emphysema development. Transgenic overexpression of MMP-9 leads to progressive lung damage in mice, whereas *Mmp9* deletion was reported to be protective during parenchymal remodeling [43,44,45]. To the contrary, other studies employing cigarette smoke-induced or chronic LPS challenge-induced emphysema models found no difference in the severity of airspace enlargement between *Mmp9* null mice and WT littermates [46,47]. Our data show that MMP-9 deficiency does not ameliorate the emphysema phenotype of βENaC-Tg mice, which is characterized by an increased lung volume, distal airspaces enlargement and destruction of alveolar septa on histology. In contrast, previous studies found that NE is an important effector molecule of these processes [6,7,9,39,48]. In these studies, genetic deletion of *NE* ameliorated the emphysematous lung morphology of βENaC-Tg mice, however, the effects on lung mechanics have not been investigated so far [9,39]. To further substantiate the differential roles of NE vs. MMP-9 in emphysema formation, we performed invasive lung function testing as a sensitive tool to compare the effects of genetic deletion of NE vs. MMP-9 on lung mechanics in βENaC-Tg mice. These measurements confirmed that lack of MMP-9 had no effect on lung function abnormalities characteristic of emphysema such as increased static compliance and inspiratory capacity of the lung, whereas genetic deletion of NE resulted in a significant improvement in these lung function parameters [9]. 

While the evaluation of the in vivo effects of genetic deletion of *Mmp9* provides important insights into the role of this protease in the complex in vivo pathogenesis of CF-like lung disease, our study also has limitations. Proteolytic activity in the lung is regulated by a complex, dynamically interconnected network of proteases and protease inhibitors, which may be different in βENaC-Tg mice and patients with CF. It is therefore plausible that the anti-protease shield may neutralize MMP-9 activity more effectively in βENaC-Tg mice, or, conversely, that NE or other proteases in this network may compensate for the lack of MMP-9 activity in mice, and that the relative contributions of proteases may differ between mice and humans. These aspects should be addressed by future studies aiming to develop therapeutic strategies targeting pulmonary proteases.

In summary, our data show that MMP-9 is not a key factor in the in vivo pathogenesis of chronic airway inflammation and structural lung damage in βENaC-Tg mice. Further, we demonstrate that NE exerts its detrimental effects on lung structure and function independently of MMP-9 in this model of CF lung disease. Finally, our results support NE rather than MMP-9 as a potential therapeutic target to reduce inflammation and structural lung damage in patients with CF.

## 4. Materials and Methods

### 4.1. Experimental Animals

All mouse experiments were approved by the Regierungspräsidium Karlsruhe (AZ 35-9185.81/G-97/14). *Mmp9*^-/-^ (B6.FVB(Cg)-*Mmp9^tm1Tvu^*/J, Jackson laboratories, Bar Habour, ME, USA) [49] were backcrossed for >6 generations and *NE*^-/-^ [50] mice were backcrossed for >12 generations to a C57BL/6 background. *Mmp9*^-/-^ and *NE*^-/-^ mice were intercrossed with βENaC-Tg mice on the C57BL/6 background [29] to generate βENaC-Tg/*Mmp9*^-/-^ and βENaC-Tg/*NE*^-/-^ mice. All experiments were performed in 6 to 8-week-old mice.

### 4.2. Genotyping

Genomic DNA was isolated from tail biopsies by incubation in 100 µL alkaline lysis buffer (25 mM NaOH and 0.2 mM EDTA; pH 12) for 40 min at 95 °C. Samples were placed on ice for 2 min and 100 µL neutralizing buffer containing 40 mM Tris-HCl at pH 5 was added. Polymerase chain reaction (PCR) was performed with supernatants using Prima Amp 2× Hot Start Red PCR mix (Steinbrenner, Wiesenbach, Germany) according to manufacturer’s instructions and primers in Table 1.

### 4.3. Bronchoalveolar Lavage (BAL) and Differential Cell Count

Mice were deeply anaesthetized via i.p. injection of 120 mg/kg ketamine (Bremer Pharma GmbH, Warburg, Germany) and 16 mg/kg xylazine (cp-pharma, Burgdorf, Germany), tracheostomized and euthanized by exsanguination. Lung lavage was performed by careful injection and aspiration of 0.0175 mL/g body weight PBS (Thermo Fisher Scientific, Darmstadt, Germany) three times using a 1 mL syringe. BAL samples were centrifuged and the cell-free supernatant was used for zymography and cytokine measurements. BAL total cell counts were determined and 3 × 10^4^ cells were adhered on a glass slide using a cytospin. Slides were stained with May-Grünwald (1:1) and Giemsa (1:10) solutions (Merck, Darmstadt, Germany) for differential cell counts. At least 400 cells were counted to obtain cell type percentages to calculate total numbers per BAL volume (cells/mL) for each cell type. 

### 4.4. Gelatin Zymography

Gelatin zymography was performed as previously described with the following modifications [51]. Fifteen μL of BAL supernatants were incubated > 10 min with 4 × Laemmli buffer (Biorad, Munich, Germany) at room temperature and run on 10% polyacrylamide gels copolymerized with 0.2% bovine type B gelatin. In βENaC-Tg mice, 10 μL BAL sample was loaded on the gel. Sodium dodecyl sulfate polyacrylamide gel electrophoresis (SDS-PAGE) was performed under non-reducing conditions at 140 V for 3–4 h at 4 °C. Next, gels were rinsed with ddH_2_O and incubated in renaturing buffer with 2.5% triton X100 (AppliChem, Darmstadt, Germany) twice for 1 h at room temperature, and subsequently incubated in development buffer containing 50 mM Tris (pH 7.5), 5 mM CaCl_2_, 1 µM ZnCl_2_ and 0.01% NaN_3_ (Sigma Aldrich, Schnelldorf, Germany) for >45 min with gentle agitation. This was followed by incubation in fresh development buffer for 38–42 h at 37 °C. Under these conditions, both zymogen and active forms of gelatinases exhibit gelatinolytic activity, leaving a clear zone that can be detected after staining. Gels were transferred to staining solution containing 5% methanol, 10% acetic acid and 0.5% Coomassie blue R-250 (Carl Roth, Karlsruhe, Germany) for 1 h. Destaining solution containing 40% methanol and 10% acetic acid (Carl Roth, Karlsruhe, Germany) was applied for 2–15 h at room temperature, until clear bands developed indicating gelatinolytic activity. Densitometric analysis was performed by FIJI software (Java 1.8.0_66). 

### 4.5. Cytokine Measurements

Interleukin (IL)-13, IL-1α, IL-1β and keratinocyte chemoattractant (KC) were measured in BAL supernatant with the cytometric bead array (CBA) (BD Biosciences, Heidelberg, Germany) and macrophage inflammatory protein (MIP)-2 concentrations were determined in BAL supernatant by ELISA (R&D Systems, Minneapolis, MN, USA) according to the manufacturer’s instructions.

### 4.6. Lung Histology

For lung volume, mean linear intercepts (MLI) and destructive index (DI) measurements lungs were inflated by applying a hydrostatic pressure of 25 cm 4% buffered formalin (Fischar, Saarbücken, Germany), ligated and fixed over night at 4 °C. Lung volume was quantified by the volume displacement measurement [52]. Subsequently, inflated lungs were sectioned, stained with hematoxylin and eosin (HE) and imaged with an Olympus IX-71 microscope (Olympus, Hamburg, Germany) using a 16× objective for MLI and DI analysis. The MLI indicating the mean linear distance between alveolar walls was determined as previously described [30,34]. Briefly, four parallel test lines were overlaid on the image, and the sum of the lengths between the intercepts of the test line with the alveolar septi was measured, divided by the total number of intercepts for all lines in 10 random fields of view per lung. The DI was used to evaluate alveolar destruction. Images were overlaid by a 45 µm × 40 µm grid of points, where points were classified as lying in a destructed or normal area according to criteria previously described [53]. The DI was determined by the ratio of the number of destructed points and total number of points. At least 300 points were evaluated for each lung. Formalin fixed non-inflated left lung lobes were used for mucus obstruction analysis. Lungs were paraffin embedded, sectioned and alcian blue periodic acid-Schiff (AB-PAS) stained. Airway mucus obstruction was assessed in the main proximal bronchi by determining the volume density, which is calculated as the ratio of the AB-PAS covering area and airway circumference [54]. 

### 4.7. Real-Time Reverse Transcriptase Quantitative Polymerase Chain Reaction (RT-qPCR)

RT-qPCR was performed using TaqMan gene expression assays for *Gob5* (Mm01320697_m1), *Muc5ac* (Mm01276718_m1), *Muc5b* (Mm00466391_m1) and *β-Actin* (Mm00607939_s1) according to manufacturer’s instructions (Applied Biosystems, Darmstadt, Germany) and analyzed as previously described [55]. 

### 4.8. Lung Function Testing

Lung mechanics testing was performed with a computer-controlled mechanical ventilator using automated breathing maneuvers as previously described [30,56]. Briefly, anesthesia was induced via i.p. injection of 80 mg/kg sodium pentobarbital (cp-pharma, Burgdorf, Germany), mice were intubated and connected to the FlexiVent ventilation apparatus (SCIREQ, Montreal, QC, Canada). Subsequently, mice were paralyzed via i.p. injection of 1.6 mg/kg pancouronium bromide (Inresa Arzneimittel GmbH, Freiburg, Germany) and continuously ventilated at a frequency of 150 breaths/min with a tidal volume of 10 mL/kg and a positive end-expiratory pressure of 3 cm H_2_O. Pressure–volume curves with stepwise increasing pressure were consecutively measured. Inspiratory capacity at a pressure of 30 cm H_2_O as well as static compliance were determined by plotting the Salazar–Knowles model to the expiratory branch of the pressure–volume loop and extracting the measurement at the linear portion of the curve by the SCIREQ (FlexiWare 5.3) software [57]. Four perturbations per mouse were measured and averaged.

### 4.9. Statistics

Statistical analyses were performed with SigmaPlot version 12.5 software (Systat Software, Erkrath, Germany) using one-way ANOVA with Fisher LSD post hoc test. Chi-square test was used for genotype distribution analysis. Data are reported as mean ± SEM. *p* < 0.05 was accepted to indicate statistical significance.

## Figures and Tables

**Figure 1 ijms-23-13405-f001:**
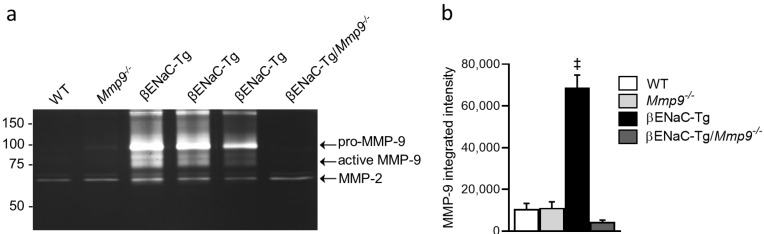
MMP-9 levels are elevated in BAL of βENaC-Tg mice. (**a**) Gelatin zymography was performed using BAL supernatant from WT, *Mmp9*^-/-^*,* βENaC-Tg or βENaC-Tg/*Mmp9*^-/-^ mice. Pro-MMP-9 (~92 kDa) and active MMP-9 (~82 and ~75 kDa) were observed in βENaC-Tg mice. Pro-MMP-2 (~72 kDa) and active MMP-2 (~65 kDa) were observed in all experimental groups. (**b**) Protein quantification of total MMP-9 by densitometry. ‡ *p* < 0.001 compared to WT, *Mmp9*^-/-^ or βENaC-Tg/*Mmp9*^-/-^. *n* = 5–8 mice per group. Statistical analysis was performed with one-way ANOVA.

**Figure 2 ijms-23-13405-f002:**
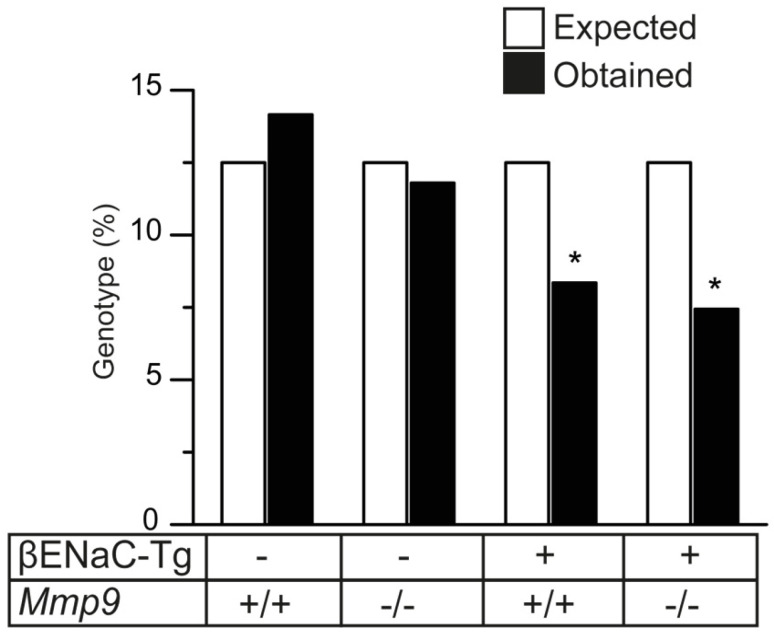
Genetic deletion of *Mmp9* does not reduce mortality in βENaC-Tg mice. Effects of genetic deletion of *Mmp9* on survival were determined by comparing the genotype frequencies expected from the Mendelian ratios with the observed genotype frequencies in 6-week-old WT, *Mmp9*^-/-^, βENaC-Tg and βENaC-Tg/*Mmp9*^-/-^ mice. * *p* < 0.05 compared to WT, *n* = 46–136 mice per genotype group. Statistical analysis was performed with chi-square test.

**Figure 3 ijms-23-13405-f003:**
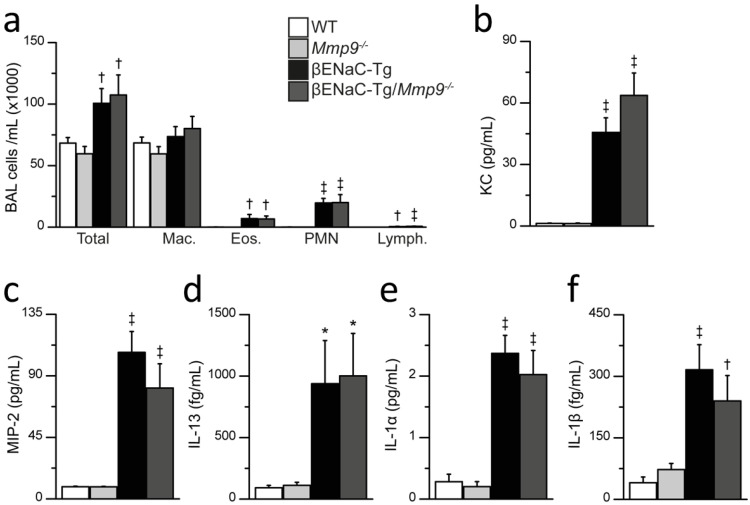
Genetic deletion of *Mmp9* does not reduce airway inflammation in βENaC-Tg mice. (**a**–**f**) Inflammatory cell counts (**a**) and cytokine levels (**b**–**f**) in BAL fluid of 6–8-week-old WT, *Mmp9*^-/-^, βENaC-Tg and βENaC-Tg/*Mmp9*^-/-^ mice. (**a**) Total and differential cell counts of bronchoalveolar macrophages (Mac.), eosinophils (Eos.), neutrophils (PMN) and lymphocyte (Lymph.) in BAL fluid. † *p* < 0.01 compared to WT or *Mmp9*^-/-^, ‡ *p* < 0.001 compared to WT or *Mmp9*^-/-^, *n* = 12–30 mice per group. (**b**–**f**) Levels of the pro-inflammatory cytokines keratinocyte chemoattractant (KC) (**b**), macrophage inflammatory protein 2 (MIP-2) (**c**), IL-13 (**d**) IL-1α (**e**) and IL-1β (**f**) in BAL supernatant. * *p* < 0.05 compared to WT or *Mmp9*^-/-^, † *p* < 0.01 compared to WT or *Mmp9*^-/-^, ‡ *p* < 0.001 compared to WT or *Mmp9*^-/-^, *n* = 10 mice per group. Statistical analysis was performed with one-way ANOVA.

**Figure 4 ijms-23-13405-f004:**
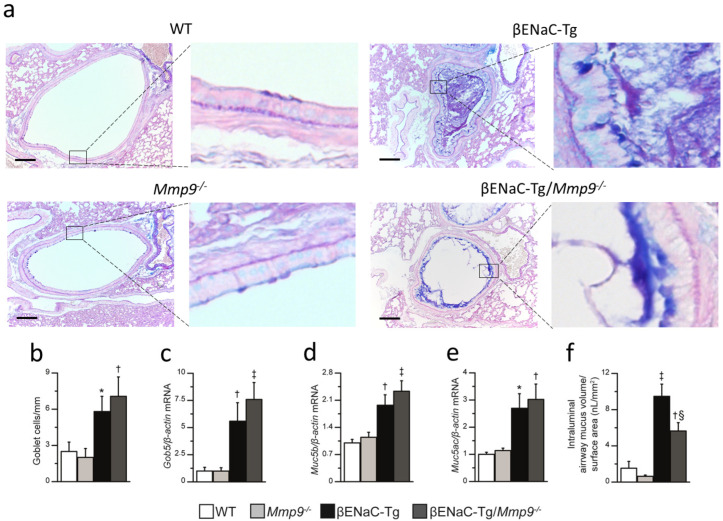
Genetic deletion of *Mmp9* does not affect goblet cell metaplasia and increased mucin expression, but partially reduces airway mucus obstruction in βENaC-Tg mice. (**a**) Representative sections of proximal main axial airway stained with alcian blue periodic acid-Schiff (AB-PAS) for assessment of goblet cell counts and airway mucus content in 6–8-week-old WT, *Mmp9*^-/-^, βENaC-Tg and βENaC-Tg/*Mmp9*^-/-^ mice. Scale bar, 100 µm. Insets (right side) show the airway epithelium, magnified from the original images using an 8× optical zoom. (**b**–**f**) Summary of goblet cell densities (**b**), mRNA expression levels of goblet cell marker *Gob5* (**c**), mucins *Muc5ac* (**d**) and *Muc5b* (**e**), and intraluminal mucus content (**f**) in WT, *Mmp9*^-/-^, βENaC-Tg and βENaC-Tg/*Mmp9*^-/-^ mice. * *p* < 0.05 compared to WT or *Mmp9*^-/-^, † *p* < 0.01 compared to WT or *Mmp9*^-/-^, ‡ *p* < 0.001 compared to WT or *Mmp9*^-/-^, § *p* < 0.01 compared to WT, *Mmp9*^-/-^ or βENaC-Tg/*Mmp9*^-/-^, *n* = 10–14 mice per group. Statistical analysis was performed with one-way ANOVA.

**Figure 5 ijms-23-13405-f005:**
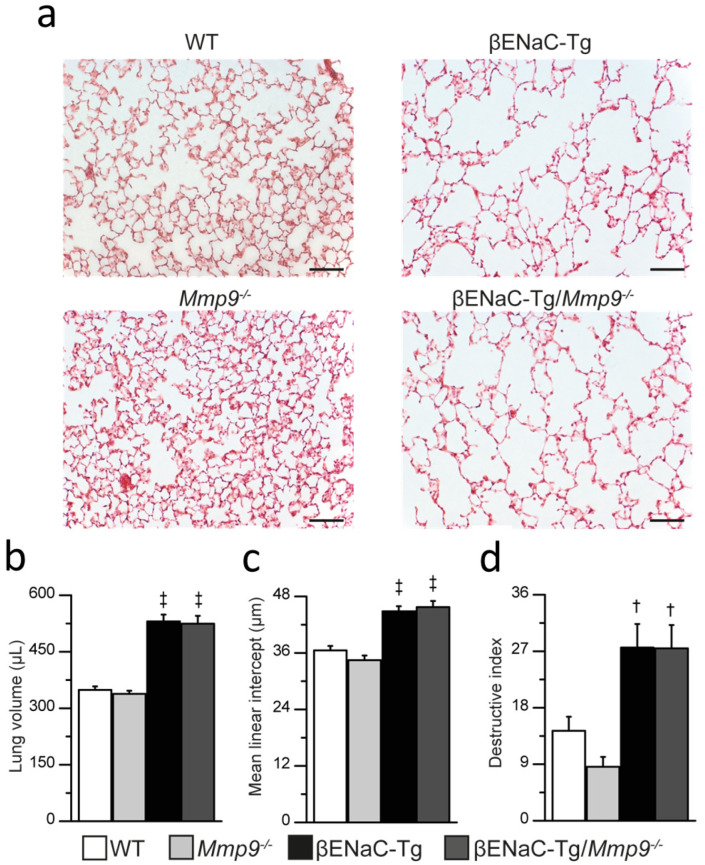
Genetic deletion of *Mmp9* does not reduce emphysema-like structural lung damage in βENaC-Tg mice. (**a**) Representative morphology of distal airspaces in hematoxylin and eosin–stained lung sections of 6–8-week-old WT, *Mmp9*^-/-^, βENaC-Tg and βENaC-Tg/*Mmp9*^-/-^ mice. Scale bar, 100 µm. (**b**–**d**) Structural lung damage was assessed by measurements of lung volume (**b**), mean linear intercepts (MLI) (**c**), and destructive index (**d**). † *p* < 0.01 compared to WT or *Mmp9*^-/-^, ‡ *p* < 0.001 compared to WT or *Mmp9*^-/-^, *n* = 9–32 mice per group. Statistical analysis was performed with one-way ANOVA.

**Figure 6 ijms-23-13405-f006:**
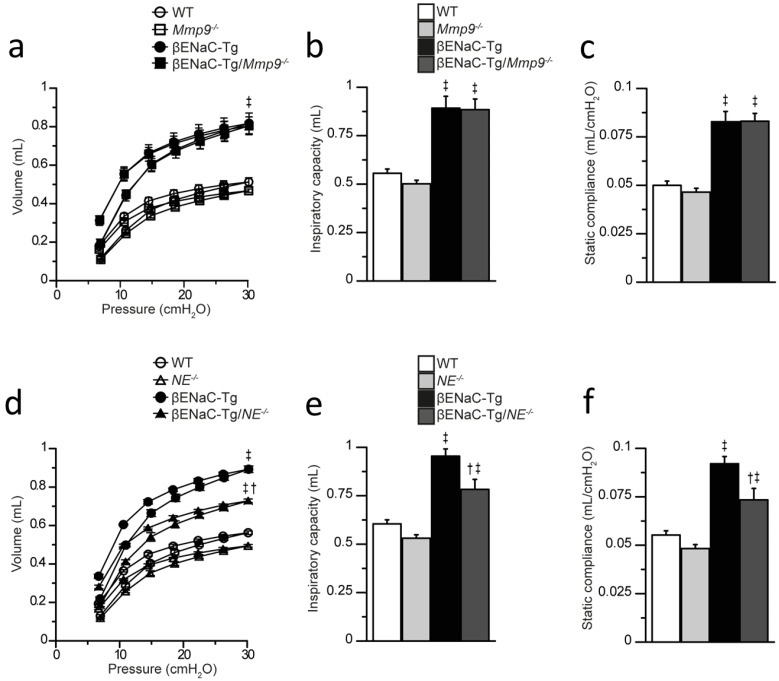
Genetic deletion of *NE*, but not *Mmp9* improves lung function in βENaC-Tg mice. (**a**–**f**) Effects of genetic deletion of *Mmp9* and *NE* on lung function impairment associated with emphysema-like lung damage in βENaC-Tg mice were determined by pressure–volume curves and measurements of the inspiratory capacity and static lung compliance in WT, *Mmp9*^-/-^, βENaC-Tg and βENaC-Tg/*Mmp9*^-/-^ mice (**a**–**c**) and WT, *NE*^-/-^, βENaC-Tg and βENaC-Tg/*NE*^-/-^ mice (**d**–**f**). † *p* < 0.01 compared to βENaC-Tg mice, ‡ *p* < 0.001 compared to WT, *NE*^-/-^
*or Mmp9*^-/-^, *n* = 8–14 mice per group. Statistical analysis was performed with one-way ANOVA.

**Table 1 ijms-23-13405-t001:** Primer sequences used for genotyping.

Gene Name	Primer Name	Sequence (5′→3′)	Annealing Temperature
*Scnn1b*	forward	CTT CCA AGA GTT CAA CTA CCG	56 °C
	reverse	TCT ACC AGC TCA GCC AGA GTG	
*Mmp9*	WT forward	GTG GGA CCA TCA TAA CAT CAC A	60 °C
	WT reverse	CTC GCG GCA AGT CTT CAG AGT A	
	KO forward	CTG AAT GAA CTG CAG GAC GA	
	KO reverse	ATA CTT TCT CGG CAG GAG CA	
*NE*	forward	GGA ACT TCG TCA TGT CAG CA	60 °C
	WT reverse	TGC ACA GAG AAG GTC TGT CG	
	KO reverse	TGG ATG TGG AAT GTG TGC GAG	

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
