# Peer review of "Genetic Deletion of Mmp9 Does Not Reduce Airway Inflammation and Structural Lung Damage in Mice with Cystic Fibrosis-like Lung Disease"

_ijms, 2022, doi:10.3390/ijms232113405_

Round 1

Reviewer 1 Report

The manuscript by Claudius et al.,Genetic deletion of Mmp9 does not reduce airway inflammation and structural lung damage in mice with cystic fibrosis- 3 like lung disease well written with proper logic & flow. Let me make some comments on the manuscript.

Major comments.

1.     In the result 2.1, authors have used gelatin zymography, but quantification is missing, authors should provide ELISA for MMP-9 from BAL sample, from that data MMP-9 quantification is possible. Zymography alone will not give any significant evidence to prove. 

2.     Authors should brief about the methods of genotype frequency calculated in methods or in results.

3.     Authors should cite proper reference of the BAL methodology or give proper small note on how counts were calculated? After centrifuge how BAL was divided into? How many images were taken in the differential count.

4.     In results 2.4, Fig (c-f) its difficult for the readers to understand. I suggest the authors to name it has qPCR results or mRNA expression results of goblet cells….. As mRNA expression is the most widely used word for gene expression studies.

5.     I have not seen any proinflammatory markers mentioned in the manuscript at least in results 2.6. Authors should provide or justify why inflammatory markers are missing in the current study. In case of NE deletion what happens to the proinflammatory cytokines? (Il-1ß, Il-6, IL-10). what will happen to NE activity? is it emphysema severity increases or decreases and how it is related to the present study. Authors should brief these key points to understand the complete mechanism. 

Minor Comments:

1.     Authors should cite recent references (last 2/3 years) in the discussion.

Reviewer 2 Report

In this study, authors show that genetic deletion of Mmp9 does not reduce airway inflammation and structural lung damage in a mouse model of cystic fibrosis. This is an interesting and well conducted study but there are some issues to be addressed prior to be considered for publication in IJMS:

Major issues

1)  Quantification of bands showed in Figure 1 should be performed to better evaluate MMP-9 levels.

2) In my opinion, measurement of TNF-α and IL17 levels in the experimental groups showed in Figure 3 could be of interest, since these cytokines are key drivers of neutrophilic inflammation, a hallmark of CF disease.  

3) Quality of images showed in Figure 4 (panel a) should be improved. Authors should also show zooms of the images showed in panel a.

Minos issues

1) Statistical analyses performed should be showed in all figure legends.

2) Authors should describe in Materials and Methods how they processed the BALF to obtain BAL supernatants for both the Gelatin Zymography and cytokine measurements.

Round 2

Reviewer 1 Report

I am happy and satisfied with the authors response, few more comments.

Comments

1.     In BAL methods “30000 cells per slide express in 104  cells per slide. The number of cell count.

2.     In the response section 5 “which we have not repeated in the current study….is it old study also MMP deletion related? If not, at least you have to mention pro-inflammatory markers in the current study and how it is related?

3.     In the response section 5 authors have told “Our focus here was on the effects of lung mechanics, … is it given any continuity statement in the discussion will be better. As study is related to emphysema/NE/MMP obviously its mechanism must be discussed in detail on all the possible aspects and if you mention future direction that will be great.

Reviewer 2 Report

Authors addressed all my commemts and sugestions in the new version of the manuscript. 

Author Response

The authors thank the Reviewer for their time and for their constructive suggestions, which highly improved this manuscript.